# EFFICIENT LEARNING OF SAFE DRIVING POLICY VIA HUMAN-AI COPILOT OPTIMIZATION

**Quanyi Li**[1]* **Zhenghao Peng**[2]* **Bolei Zhou**[3]
[1]Centre for Perceptual and Interactive Intelligence, [2]The Chinese University of Hong Kong,
[3]University of California, Los Angeles

## ABSTRACT

Human intervention is an effective way to inject human knowledge into the loop of reinforcement learning, bringing fast learning and training safety. But given the very limited budget of human intervention, it is challenging to design when and how human expert interacts with the learning agent in the training. In this work, we develop a novel human-in-the-loop learning method called Human-AI Copilot Optimization (HACO). To allow the agent's sufficient exploration in the risky environments while ensuring the training safety, the human expert can take over the control and demonstrate to the agent how to avoid probably dangerous situations or trivial behaviors. The proposed HACO then effectively utilizes the data collected both from the trial-and-error exploration and human's partial demonstration to train a high-performing agent. HACO extracts proxy state-action values from partial human demonstration and optimizes the agent to improve the proxy values while reducing the human interventions. No environmental reward is required in HACO. The experiments show that HACO achieves a substantially high sample efficiency in the safe driving benchmark. It can train agents to drive in unseen traffic scenes with a handful of human intervention budget and achieve high safety and generalizability, outperforming both reinforcement learning and imitation learning baselines with a large margin. Code and demo videos are available at: `https://decisionforce.github.io/HACO/`.

## 1 INTRODUCTION

How to effectively inject human knowledge into the learning process is one of the key challenges to training reliable autonomous agents in safety-critical applications. In reinforcement learning (RL), researchers can inject their intentions into the carefully designed reward function. The learning agent freely explores the environment to collect the data and develops the desired behaviors induced by the reward function. However, RL methods bear two drawbacks that limit their applications in safety-critical tasks: First, the nature of trial-and-error exploration exposes RL agent to dangerous situations (Saunders et al., 2017). Second, it is difficult to summarize all the intended behaviors to be learned into the reward function. Taking the driving vehicle as an example, an ideal policy should obtain a set of skills, such as overtaking, yielding, emergent stopping, and negotiation with other vehicles. It is intractable to manually design a reward function that leads to the emergence of all those behaviors in the trained agent. To mitigate these two challenges, practitioners enforce the human intentions through imitation learning (IL) where the agent is trained to imitate the expert-generated state and action sequences. During the demonstration, the premature agent does not interact with the risky environment and thus the training safety is ensured. High-quality expert demonstrations provide direct the optimal solution for the agent to imitate from. However, IL paradigm suffers from the distributional shift problem (Ross & Bagnell, 2010; Ross et al., 2011) while the induced skills are not sufficiently robust with respect to changes in the control task (Camacho & Michie, 1995).

Different from vanilla RL or IL, human-in-the-loop learning is an alternative paradigm to inject human knowledge, where a human subject accompanies the agent and oversees its learning process. Previous works require the human to either passively advise which action is good (Mandel et al., 2017) or evaluate the collected trajectories (Christiano et al., 2017; Guan et al., 2021; Reddy et al.,

---

*Quanyi Li and Zhenghao Peng contribute equally to this work.

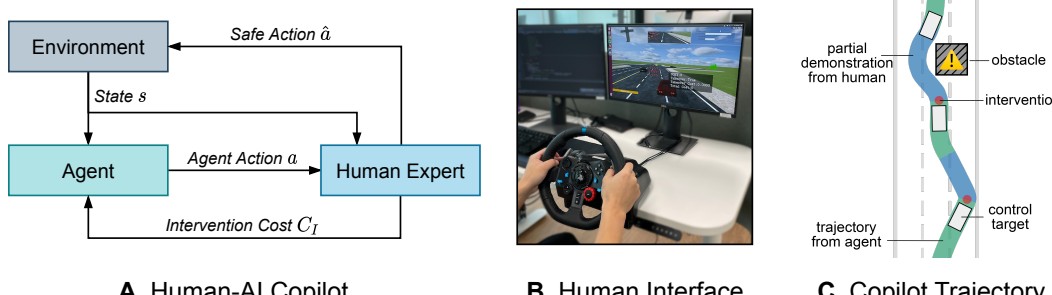

**A**. Human-AI Copilot      **B**. Human Interface      **C**. Copilot Trajectory

Figure 1: **A.** In the proposed human-in-the-loop method, human subjects will decide whether to intervene in the execution of the learning agent and demonstrate the correct action. Such intervention incurs a cost to the agent such that it learns to minimize the total cost during the training. **B.** The human interface for copilot during the training. The human subject can steer the wheel or press the paddle to start the intervention and then the demonstration data will be recorded. **C.** An illustrative driving trajectory where the green segments are generated by the agent and the blue segments are the demonstrations of the human expert to overcome the dangerous situations.

2018; Warnell et al., 2018; Christiano et al., 2017; Sadigh et al., 2017; Palan et al., 2019). This kind of passive human involvement exposes the human-AI system to risks since the agent explores the environment without protection. Some other works require the human to merely intervene in the exploration by terminating the episode (Saunders et al., 2018; Zhang & Cho, 2016), but it is not practical to terminate and reset the environment instantly in the real world (Xu et al., 2020). Intervening and taking over the control from the learning agent is a natural approach to safeguard the human-AI system (Kelly et al., 2019; Spencer et al., 2020). However, a challenge exhibited in previous works is the budget of human intervention. Since human cognitive resource is precious and limited, it is essential to carefully design when and how the human expert involves in the learning process so that the human knowledge can be injected effectively.

In this work, we propose an efficient human-in-the-loop learning method called ***Human-AI Copilot Optimization (HACO)***. The key feature of HACO is that it can learn to minimize the human intervention and adjust the level of automation to the learning agent adaptively during the training. As shown in Figure 1 A, HACO allows the human expert to take over the human-AI system in a proactive manner. If the human decides to intervene in the action of the agent, he/she should demonstrate the correct actions to overcome current undesired situations to the learning agent. The human intervention and the partial demonstration are two sources of informative training data. We use offline RL technique to maintain a proxy value function of the human-AI mixed behavior policy even though the agent doesn't have the access to the environmental reward during training. To encourage the exploration in the state-action space permitted by human, we also maximize the entropy of action distribution of the agent if the agent is not taken over.

Experiments in the virtual driving environments MetaDrive (Li et al., 2021) and CARLA (Dosovitskiy et al., 2017) show that, with an economic human budget, HACO outperforms RL and IL baselines with a substantial margin in terms of sample efficiency, performance, safety, and generalizability in the unseen testing environment. Thus the human-AI copilot optimization is an efficient learning paradigm to inject human knowledge in an online setting.

## 2    RELATED WORK

**Learning from Demonstration.** Passive imitation learning such as behavior cloning (Widrow, 1964; Osa et al., 2018; Huang et al., 2020; Sun et al., 2020) and recently proposed offline RL methods (Kumar et al., 2020; Fujimoto et al., 2018; Wu et al., 2019) train agents from an out-of-the-shelf data set and guarantee the training safety, since no interaction with the environment is needed. Inverse RL methods (Ng et al., 2000; Abbeel & Ng, 2004; Fu et al., 2017; Bloem & Bambos, 2014) learn a reward function from the human demonstration and then use it to incentivize the agents to master the intended behaviors. Proposed more recently, GAIL (Ho & Ermon, 2016) and its variants (Song et al., 2018; Sasaki et al., 2018; Kostrikov et al., 2018) and SQIL (Reddy et al., 2019) compare the trajectory similarity between agents and humans and thus require the agent to interact with the environment. Similar to RL methods, this paradigm exposes the agent to potentially dangerous situations.

**Human-in-the-loop Learning Methods.** Many works focus on incorporating human in the training loop of conventional RL or IL paradigms. DAgger (Ross et al., 2011) and its extended methods (Kelly et al., 2019; Zhang & Cho, 2016; Hoque et al., 2021) correct the compounding error (Ross & Bagnell, 2010) of behavior cloning by periodically requesting expert to provide more demonstration. Instead of proving demonstration upon requests, Human-Gated DAgger (HG-DAgger) (Kelly et al., 2019), Expert Intervention Learning (EIL) (Spencer et al., 2020) and Intervention Weighted Regression (IWR) (Mandlekar et al., 2020) empower the expert to intervene exploration and carry the agent to safe states. However, these methods do not impose constraints to reduce human intervention and do not utilize the data from the free exploration of the agent. Human subjects can also involve in the loop providing preferences based on evaluative feedback on two behavior sequences generated by the agent (Christiano et al., 2017; Sadigh et al., 2017; Palan et al., 2019; Ibarz et al., 2018; Cui & Niekum, 2018).

Human-AI copilot or shared autonomy is a more intimate form of the human-in-the-loop methods. The AI agent and human are working together simultaneously to achieve a common goal. By giving human guidance and feedback at run-time instantly, the explorable state and action spaces can be greatly narrowed down (Saunders et al., 2018). The learning goal can further match the task objective by providing extra human feedback combined with reward function (Reddy et al., 2018; Warnell et al., 2018; Wu et al., 2021; Cederborg et al., 2015; Arumugam et al., 2019). Human-AI copilot is helpful and practical when applying RL to real world tasks where safety constraints must be satisfied (García & Fernández, 2015; Amodei et al., 2016; Bharadhwaj et al., 2020; Alshiekh et al., 2018). In our previous work (Peng et al., 2021), we made attempt to develop a method called Expert-Guided Policy Optimization (EGPO) where a PPO expert policy is involved to monitor the learning agent. The difference can be summarized as twofold: (1) We substitute the expert with human and design special mechanism to mitigate the delayed feedback error; (2) Based on the comprehensive ablation study and prototyping, we remove redundant designs like takeover function and the need of reward function, making the proposed method simple yet effective.

Reducing human burden is a major challenge in human-in-the-loop methods. A feasible solution is to learn an intervention function that imitates human intervention signals and stops the catastrophic actions of agents (Kelly et al., 2019; Zhang & Cho, 2016; Saunders et al., 2017; Abel et al., 2017), which can relieve the mental stress of the human subject during training. In this work, we devise our learning scheme explicitly to include the human cognitive cost as one of the objectives to minimize.

## 3 HUMAN-AI COPILOT OPTIMIZATION

In this section, we introduce *Human-AI Copilot Optimization (HACO)*, an efficient learning algorithm that trains agents from human interventions, partial demonstrations and free exploration. For human-in-the-loop learning, it is essential to design when and how to engage human subjects. The major issue is the *cognitive cost* of the human subject (Zhang et al., 2021). Frequent querying might bring tremendous cognitive cost and exhaust the human expert, causing incorrect or delayed feedback that hinders the training. Thus the proposed pipeline aims to minimize the human intervention cost during the training, which reduces the reliance on the expert's demonstration over time and improves the learning agent's autonomy. The overall workflow of HACO is presented in Algorithm 1.

### 3.1 HUMAN-AI COPILOT TRAINING PARADIGM

We aim to learn an autonomous agent with policy $\pi_n(a_n|s)$ that can make informed action $a_n$ in state $s$. As shown in Fig. 1, we frame the *human-AI copilot paradigm* that extends the standard reinforcement learning diagram by incorporating a human expert. At each step, the human expert oversees current state and decides whether to intervene. If necessary, he/she will execute human action $a_h$ to overwrite the agent's action $a_n$. We denote the human intervention by a Boolean indicator $I(s, a_n)$ and thus the action applied to the environment is called the safe action $\hat{a} = I(s, a_n)a_h + (1 - I(s, a_n))a_n$. Denoting the human policy as $\pi_h$, the actual trajectories occurred during training are derived from a shared behavior policy $\pi_b$:

$$\pi_b(a|s) = \pi_n(a|s)(1 - I(s, a)) + \pi_h(a|s)G(s), \tag{1}$$

wherein $G(s) = \int_{a' \in \mathcal{A}} I(s, a')\pi_n(a'|s)da'$ is the probability of the agent choosing an action that will be rejected by the human.

We call the transition sequences during the takeover $\{(s_t, a_{n,t}, a_{h,t}, I(s_t, a_{n,t}), s_{t+1}), ...\}$ as the *partial demonstration*. The partial demonstration and the free exploration transitions will be recorded in the replay buffer $\mathcal{B}$ and fed to the training pipeline. Note that we do not require to store environmental reward and cost into the buffer since the proposed method does not need them.

In the human-AI copilot training, the human is obligated to guide the agent learning and safeguard the learning process by proactively taking over the control if necessary. This paradigm rules out the dispensable states and mitigates the safety concern in free exploration of RL and active imitation learning methods (Ross et al., 2011). Different from previous offline RL works training from fixed dataset (Bojarski et al., 2016; Ho & Ermon, 2016; Reddy et al., 2019; Kumar et al., 2020; Fujimoto et al., 2018; Wu et al., 2019) where no closed loop feedback is accessible, the human-AI copilot training produces *partial demonstrations* that contains the necessary human knowledge to overcome dangerous situations into the learning. The copilot nature alleviates the distributional shift problem, since the human intervenes when the agent performs suspicious behaviors, so that there is a continuity of the state visitation between the agent and the expert.

In next section, we will introduce how we instantiate the human-AI copilot paradigm with a human-efficient algorithm that can effectively optimize the agent toward safe and high-performing policy.

## 3.2 Learning Objectives

We form three objectives that fully utilize the human data: (1) Agent should maximize a *proxy value function* $Q(s, a)$ which reflects human intentions on how to finish the task. (2) Agent should explore thoroughly to visit the state-action subspace permitted by the human. Concretely, we maximize the action distribution entropy $\mathcal{H}(\pi(\cdot|s))$. (3) Agent should maximize the level of automation and reduce human intervention. Episodic human intervention is estimated by an intervention value function $Q^I(s, a)$ based on the step-wise intervention cost $C(s, a)$. Thus the overall learning objective of HACO becomes:

$$\max_\pi \mathbb{E}[Q(s, a) + \mathcal{H}(\pi) - Q^I(s, a)]. \tag{2}$$

We then discuss the practical implementation of aforementioned design goals.

**Proxy value function.** HACO follows reward-free setting so we can't estimate the expected state-action value based on a ground-truth reward function defined by the environment. We instead estimate a proxy value function $Q(s, a; \phi)$ ($\phi$ is model parameters) that captures the ordinal preference of human experts, which implicitly reflects human intentions. We utilize the conservative Q-learning (Kumar et al., 2020) and form the optimization problem of the proxy value function as:

$$\min_\phi \mathbb{E}_{(s, a_n, a_h, I(s, a_n)) \sim \mathcal{B}} [I(s, a_n)(Q(s, a_n; \phi) - Q(s, a_h; \phi))]. \tag{3}$$

The above optimization objective can be interpreted as being optimistic to the human's action $a_h$ and pessimistic to the agent's action $a_n$. The proxy value function learns to represent the high-value state-action subspace preferred by the human expert.

**Entropy regularization.** If the learning agent visits human-preferable subspace insufficiently during free explorable sampling, the states evoking high proxy value are rarely encountered, making the back-propagation of the proxy value to preceding states difficult and thus damaging the learning. To encourage exploration, we adopt the entropy regularization technique in (Haarnoja et al., 2018) and forms auxiliary signal to update the proxy value function apart from Eq. 3:

$$\min_\phi \mathbb{E}_{(s_t, \hat{a}_t, s_{t+1}) \sim \mathcal{B}} [y - Q(s_t, \hat{a}_t; \phi)]^2, \quad y = \gamma \mathbb{E}_{a' \sim \pi_n(\cdot|s_{t+1})} [Q(s_{t+1}, a'; \phi') - \alpha \log \pi_n(a'|s_{t+1})], \tag{4}$$

wherein $\hat{a}_t$ is the executed action at state $s_t$, $\phi'$ denotes the delay updated parameter of the target network, $\gamma$ is the discount factor. Since the environment reward is not accessible to HACO, we remove the reward term in the update target $y$. Combining Eq. 3 and Eq. 4, the formal optimization objective of the proxy value function becomes:

$$\min_\phi \mathbb{E}_{\mathcal{B}} [(y - Q(s_t, \hat{a}_t; \phi))^2 + I(s_t, a_{n,t})(Q(s_t, a_{n,t}; \phi) - Q(s_t, a_{h,t}; \phi))]. \tag{5}$$

---

**Algorithm 1:** The workflow of HACO during training

---

1  Initialize an empty replay buffer $\mathcal{B}$
2  **while** *Training is not finished* **do**
3     **while** *Episode is not terminated* **do**
4        $a_{n,t} \sim \pi_n(\cdot|s_t)$ Retrieve agent's action
5        $I(s_t, a_{n,t}) \leftarrow$ Human expert decides whether to intervene by observing current state $s_t$
6        **if** $I(s_t, a_{n,t})$ *is True* **then**
7           $a_{h,t} \leftarrow \pi_h(\cdot|s_t)$ Retrieve human's action
8           Apply $a_{h,t}$ to the environment
9        **else**
10          Apply $a_{n,t}$ to the environment
11       **if** $I(s_t, a_{n,t})$ *is True and* $I(s_{t-1}, a_{n,t-1})$ *is False* **then**
12          $C(s_t, a_{n,t}) \leftarrow$ Compute intervention cost following Eq. 6
13       **else**
14          $C(s_t, a_{n,t}) \leftarrow 0$ Set intervention cost to zero
15       Record $s_t, a_{n,t}, I(s_t, a_{n,t})$ and $a_{h,t}$ (if $I(s_t, a_{n,t})$) to the buffer $\mathcal{B}$
16    Update proxy value $Q$, intervention value $Q^I$ and policy $\pi$ according to Eq. 5, Eq. 7, Eq. 8 respectively

---

**Reducing human interventions.** Directly optimizing the agent policy according to the proxy value function will lead to failure when evaluating the agent without human participation. This is because $Q(s, a)$ represents the proxy value of the mixed behavior policy $\pi_b$ instead of the learning agent's $\pi_n$ due to the existence of human intervention. It is possible that the agent learns to deliberately abuse human intervention by always taking actions that violate human intentions, such as driving off the road when near the boundary, which forces human to take over and provide demonstrations. In this case, the level of automation for the agent is low and the human subject exhausts to provide demonstrations. Ablation study result in Table 2(c) illustrates this phenomenon.

To economically utilize human budget and reduce the human interventions over time, we punish the agent action that triggers human intervention in a mild manner by using the cosine similarity between agent's action and human's action as the intervention cost function in the form below:

$$C(s, a_n) = 1 - \frac{a_n^{\mathsf{T}} a_h}{||a_n||||a_h||}, \quad a_h \sim \pi_h(\cdot|s). \tag{6}$$

The agent will receive large penalty only when its action is significantly different from the expert action in terms of cosine similarity.

A straightforward form of $C$ is a constant $+1$ when human expert issues intervention. However, we find that there usually exists temporal mismatch $\epsilon$ between human intervention and faulty actions so that the intervention cost is given to the agent at a delayed time step $t + \epsilon$. It is possible that the agent's action $a_{n,t+\epsilon}$ is a correct action that saves the agent itself from dangers but is mistakenly marked as faulty action that triggers human intervention. In the ablation study, we find that using the constant cost raises inferior performance compared to the cosine similarity.

As shown in Line 11-14 of Algorithm 1, we only yield non-zero intervention cost at the first step of human intervention. This is because the human intervention triggered by the exact action $a_{n,t}$ indicates this action violates the underlying intention of human at this moment. Minimizing the chance of those actions will increase the level of automation.

To improve the level of automation, we form an additional intervention value function $Q^I(s, a)$ as the expected cumulative intervention cost, similar to estimating the state-action value in Q-learning through Bellman equation:

$$Q^I(s_t, a_{n,t}) = C(s_t, a_{n,t}) + \gamma \mathop{\mathbb{E}}_{s_{t+1} \sim \mathcal{B}, a_{t+1} \sim \pi_n(\cdot|s_{t+1})} [Q^I(s_{t+1}, a_{t+1})]. \tag{7}$$

This value function is used to directly optimize the policy.

**Learning policy.** Using the entropy-regularized proxy value function $Q(s, a)$ as well as the intervention value function $Q^I(s, a)$, we form the the policy improvement objective as:

$$\max_{\theta} \mathop{\mathbb{E}}_{s_t \sim \mathcal{B}} [Q(s_t, a_n) - \alpha \log \pi_n(a_n|s_t; \theta) - Q^I(s_t, a_n)], \quad a_n \sim \pi_n(\cdot|s_t; \theta). \tag{8}$$

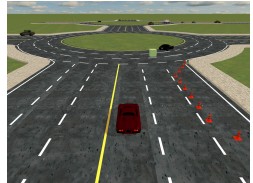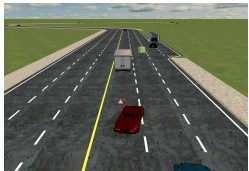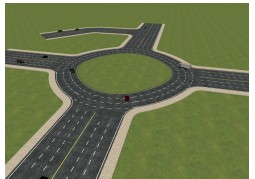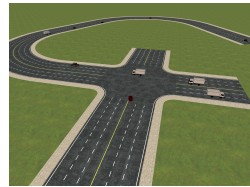

Figure 2: Examples of the safe driving environments from MetaDrive used in the experiments.

# 4 EXPERIMENTS

## 4.1 EXPERIMENTAL SETTINGS

**Task.** We focus on the driving task in this work. This is because driving is an important decision making problem with a huge social impact, where safety and training efficiency are critical. Since many researches on autonomous driving employ human in a real vehicle (Bojarski et al., 2016; Kelly et al., 2019), the human safety and human cognitive cost become practical challenges that limit the application of learning-based methods in industries. Therefore, the driving task is an ideal benchmark for the human-AI copilot paradigm.

**Simulator.** Considering the potential risks of employing human subjects in physical experiments, we benchmark different approaches in the driving simulator. We employ a lightweight driving simulator MetaDrive (Li et al., 2021), which preserves the capacity to evaluate the safety and generalizability in unseen environments. The simulator is implemented based on Panda3D (Goslin & Mine, 2004) and Bullet Engine that has high efficiency as well as accurate physics-based 3D kinetics. MetaDrive uses procedural generation to synthesize an unlimited number of driving maps for the split of training and test sets, which is useful to benchmark the generalization capability of different approaches in the context of safe driving. Some generated driving scenes are presented in Fig. 2. The simulator is also extremely efficient and flexible so that we can run the human-AI copilot experiment in real-time. Though we mainly describe the setting of MetaDrive in this section, we also experiment on CARLA (Dosovitskiy et al., 2017) simulator in Sec. 4.3.

**Training Environment.** In the simulator, the task for the agent is to steer the target vehicle with low-level control signal, namely acceleration, brake and steering, to reach the predefined destination and receive a *success* flag. The ratio of episodes where the agent successfully reaches the destination is called the *success rate*. To increase the difficulty of the task, we scatter obstacles randomly in each driving scene such as movable traffic vehicles, fixed traffic cones, and warning triangles.

The observation contains (1) the current states such as the steering, heading, velocity and relative distance to boundaries *etc.*, (2) the navigation information that guides the vehicle toward the destination, and (3) the surrounding information encoded by a vector of 240 Lidar-like distance measures of the nearby vehicles.

Though HACO does not receive environmental reward during training, we provide reward function to train baseline methods and evaluate HACO in test time. The reward function contains a dense driving reward, speed reward and a sparse terminal reward. The driving reward measures the longitudinal movement toward destination. We also reward agent according to its velocity and give the sparse reward $+20$ when the agent arrives at the destination.

Each collision to the traffic vehicles or obstacles yields $+1$ environmental cost. Note that HACO can not access this cost during the training. This cost is used to train safe RL baselines as well as for testing the safety of trained policies. We term the episodic cost as *safety violation* which is the measurement on the safety of a policy.

We invite the human expert to supervise the real-time exploration of the learning agent with hands on the steering wheel, as shown in the Fig. 1B. When a dangerous situation is going to happen, the human takes over the vehicle by pressing the paddle besides the wheel and starts controlling the vehicle by steering the wheel and stepping the pedals.

**Split of training and test sets.** Different from the conventional RL setting where the agent is trained and tested in the same fixed environment, we focus on evaluating the generalization performance through testing the trained agents in separated test environments. We split the driving scenes into the training set and test set with 50 different scenes in each set. After each training iteration, we roll out the learning agent *without guardian* in the test environments and record success rate and safety violation given by the environment and present it in Table 1.

Table 1: The performance of different approaches in safe driving benchmark.

| Category | Method | Total Training Safety Violation | Training Data Usage | Test Return | Test Safety Violation | Test Success Rate |
|---|---|---|---|---|---|---|
| *Expert* | *Human* | - | - | *358.19* $\pm_{86.00}$ | *0.16* $\pm_{0.61}$ | *0.98* |
| RL | SAC-RS | 2.76K $\pm$ 0.95K | 1M | **386.77** $\pm_{35.1}$ | 0.73 $\pm_{1.18}$ | 0.82 $\pm_{0.18}$ |
|  | PPO-RS | 24.34K $\pm_{3.56K}$ | 1M | 335.39 $\pm_{12.41}$ | 3.41 $\pm_{1.11}$ | 0.69$\pm_{0.08}$ |
| Safe RL | SAC-Lag | 1.84K $\pm$ 0.49K | 1M | 351.96 $\pm_{101.88}$ | **0.72** $\pm_{0.49}$ | 0.73 $\pm_{0.29}$ |
|  | PPO-Lag | 11.64K $\pm$ 4.16K | 1M | 299.99 $\pm_{49.46}$ | 1.18 $\pm_{0.83}$ | 0.51 $\pm_{0.17}$ |
|  | CPO | 4.36K $\pm_{2.22K}$ | 1M | 194.06 $\pm_{108.86}$ | 1.71 $\pm_{1.02}$ | 0.21 $\pm_{0.29}$ |
| Offline RL | CQL | - | 36K | 156.4 $\pm_{31.94}$ | 6.82 $\pm_{5.1}$ | 0.11 $\pm_{0.07}$ |
| IL | BC | - | 36K | 101.63 $\pm_{16.06}$ | 1.00 $\pm_{0.45}$ | 0.01 $\pm_{0.03}$ |
|  | GAIL | 3.70K $\pm_{2.43K}$ | 36K | 136.08 $\pm_{18.73}$ | 4.42 $\pm_{1.89}$ | 0.13 $\pm_{0.03}$ |
| Human-in-the-loop | HG-DAgger | 38.35 | 50K | 111.87 | 2.38 | 0.04 |
|  | IWR | 77.38 | 50K | 299.78 | 3.39 | 0.64 |
| Ours | **HACO** | **30.14** $\pm$ 11.36 | **30K**[*] | 349.25 $\pm$ 11.45 | 0.79 $\pm$ 0.31 | **0.83** $\pm$ 0.04 |

[*] During HACO training, in 8316 $\pm$ 497.90 steps out of the total 30K steps the human expert intervenes and overwrites the agent's actions. The whole training takes about 50 minutes.

**Implementation details.** We conduct experiments on the driving simulator and implement algorithms using RLLib (Liang et al., 2018), an efficient distributed learning system. When training the baselines, we host 8 concurrent trials in an Nvidia GeForce RTX 2080 Ti GPU. Each trial consumes 2 CPUs with 8 parallel rollout workers. Except human-in-the-loop experiments, all baseline experiments are repeated 5 times with different random seeds. The main experiments of HACO is conducted on a local computer with an Nvidia GeForce RTX 2070 and repeat 3 times. The ablations and baseline human-in-the-loop experiments repeat once due to the limited human budget. One human subject participates in each experiment. In all tables and figures, we provide the standard deviation if the experiments are repeated multiple runs with different random seeds. Information about other hyper-parameters is given in the Appendix.

## 4.2 BASELINE COMPARISON

We compare our method to vanilla RL and Safe RL methods which inject the human intention and constraint through the pre-defined reward function and cost function. We test native RL methods, PPO (Schulman et al., 2017) and SAC (Haarnoja et al., 2018), with cost added to the reward as auxiliary negative reward, called reward shaping (RS). Three common safe RL baselines Constraint Policy Optimization (CPO) (Achiam et al., 2017), PPO-Lagrangian (Stooke et al., 2020), SAC-Lagrangian (Ha et al., 2020) are evaluated.

Apart from the RL methods, we also generate a human demonstration dataset containing one-hour expert's demonstrations where there are about 36K transitions in the training environments. For the high-quality demonstrations in the dataset, the success rate of the episodes reaches 98% and the safety violation is down to 0.16. Using this dataset, we evaluate passive IL method Behavior Cloning, active IL method GAIL (Ho & Ermon, 2016) and offline RL method CQL (Kumar et al., 2020). We also run the Human-Gated DAgger (HG-DAgger) (Kelly et al., 2019) and Intervention Weighted Regression (IWR) (Mandlekar et al., 2020) as the baselines of human-in-the-loop methods based on this dataset and the human-AI copilot workflow.

**Training-time Safety.** The training-time safety is measured by the *total training safety violation*, the total number of critical failures occurring in the training. Note that the environmental cost here is different from the human intervention cost in HACO. As illustrated in Table 1 and Fig. 3**A**, HACO achieves huge success in training time safety. Apart from the empirical results, we provide proof to show the training safety can be bound by the guardian in Appendix. Under the protection of the human expert, HACO yields only 30.14 total safety violations in the whole training process, two orders of magnitude better than other RL baselines, even though HACO does not access the environmental cost. IWR and HG-DAgger also achieve drastically lower training safety violations, showing the power of human-in-the-loop methods. The most competitive RL baseline SAC-RS, which achieves similar test success rate, causes averagely 2767.77 training safety violations which are much higher

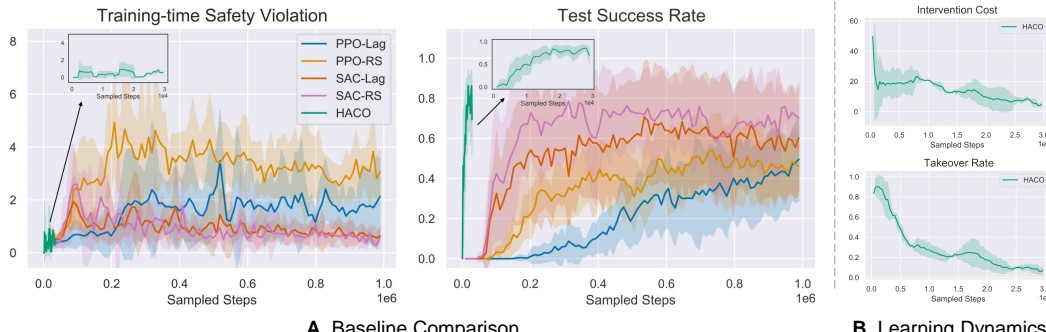

Figure 3: **A**. The performance of RL methods and HACO. HACO achieves superior training time safety and test success rate. **B**. Dynamics of human cognitive cost. The takeover rate is the proportion of the human takeover steps in an episode. In the training process, the takeover rate and the episodic intervention cost gradually reduce and the agent obtains higher level of autonomy.

than HACO. The active IL method GAIL also has significantly higher safety violations than HACO and its performance is unsatisfactory.

From the perspective of safety, we find that the reward shaping technique is inferior compared to the Lagrangian method, both for SAC and PPO variants. PPO causes more violations than SAC, probably due to the relatively lower sample efficiency and slower convergence speed.

**Sample Efficiency and Human Cognitive Cost.** The human-AI system is not only protected so well by the human, but achieves superior sample efficiency with limited data usage. As shown in Fig. 3**A** and Table 1, we find that HACO is an order of magnitude more efficient than RL baselines. HACO achieves 0.83 test success rate by merely interacting with the environment in the 30K steps, wherein only averagely 8,316 steps the human provides safe actions as demonstration. During nearly 50 minutes of human-AI copilot, there are only 27% steps that the human provides demonstrations.

Human-in-the-loop baselines IWR and HG-DAgger consume 50K steps of human budget and only IWR can achieve satisfactory success rate. By prioritizing samples from human intervention, IWR manages to learn key actions from human intervention to escape dangerous situations caused by the compounding error. Without re-weighting the human takeover data, HG-Dagger fails to learn from a few but important human demonstrations. The learning curves of these two methods can be found in the Appendix.

Unlike the success of HACO, all the learning-from-demonstration methods fail with the dataset containing 36K transitions. Compared to IL methods which optimize agents to imitate exact actions at each time step, HACO considers the learning on the trajectory basis. We incentivize the agent to choose an action that can bring potential return in future trajectory, instead of only mimicking the expert's behaviors at each step. On the other hand, HACO gathers expert data in an online manner through human-AI copilot, which better mitigates the distributional shift severe in offline RL methods.

**Learning Dynamics.** The intervention minimization mechanism in HACO reduces human cognitive cost. As shown in Fig. 3**B**, the takeover rate gradually decreases in the course of learning. The curve of episodic intervention cost suggests that the human intervention frequency becomes lower and the similarity between agent's action and human's action increases. We also provide visualization of the learned proxy value function in the Appendix, showing that the learning scheme of HACO can effectively encode human preference into the proxy values.

### 4.3 ABLATION STUDY

**Takeover Policy Analysis.** We request the human subjects to try two intervention strategies. The first is to take over in a low frequency and produce a long trajectory at each intervention. In this way the intervention cost becomes sparse. The other strategy is to intervene more frequently and provide fragmented demonstrations. In Table 2(a), the experiment shows that the proposed HACO works better with dense human intervention signals. Agent trained with long trajectories achieves inferior success rate and episodic reward than agents trained with dense intervention signals.

Table 2: The test performance when ablating components in HACO. All experiments here are conducted by human subjects. Some experiments only repeat once due to limited human budget.

| Experiment | Test Return | Test Cost | Test Success Rate |
|---|---|---|---|
| **(a)** Human intervenes less frequently | 220.03 | 0.867 | 0.397 |
| **(b)** W/o cosine similarity intervention cost | 23.23 | 0.42 | 0.00 |
| **(c)** W/o intervention minimization | 84.77 | 1.07 | 0.00 |
| **HACO** | $349.25_{\pm 11.45}$ | $0.79_{\pm 0.31}$ | $0.83_{\pm 0.04}$ |

Table 3: Comparison between HACO and PPO. Agents are trained in CARLA town 1 and tested in the unseen CARLA town 2.

| Algorithm | Test Safety Violation | Test Return | Test Success Rate | Train Samples |
|---|---|---|---|---|
| PPO | 80.84 | 1591.00 | 0.35 | 500,000 |
| HACO | 11.84 | 1579.03 | 0.35 | 8,000 |

**Cosine Similarity Cost Function.** As shown in Table 2(b), we replace the intervention cost function in Eq. 6 to a constant value $+1$ if human intervention happens. We find the agent learns to stay in the spawn points and does not move at all in test time. As discussed in Sec. 3.2, it is possible that the human intervenes in incorrect timing. This makes agent fail to identify how to drive correctly. Using the negative cosine similarity to measure the divergence between agent and human's actions alleviates this phenomenon since the human intervention penalty is down-weighted when the agent provides action that adheres human intention.

**Intervention Minimization.** As shown in Table 2(c), when removing the intervention minimization mechanism, the agent drives directly toward the boundary. This is because the agent learns to abuse human expert to take over all the time, which increases proxy values but causes consistent out-of-the-road failures in testing. This result shows the importance of intervention minimization.

**CARLA Experiment.** To test the generality of HACO, we run HACO in the CARLA simulator (Dosovitskiy et al., 2017). We use the top-down semantic view provided by CARLA as the input and a 3-layer CNN as the feature extractor for HACO and the PPO baseline. For PPO, the reward follows the setting described in CARLA and is based on the velocity and the completion of the road. We train HACO (with a human expert) and PPO in CARLA town 1 and report the test performance in CARLA town 2. Table 3 shows that the proposed HACO can be successfully deployed in the CARLA simulator with visual observation and achieve comparable results. Also, it can train the driving agent with a new CNN feature-extractor in 10 minutes with only 8,000 samples in the environment. The video is available at: `https://decisionforce.github.io/HACO/`.

## 5 CONCLUSION

We develop an efficient human-in-the-loop learning method, *Human-AI Copilot Optimization (HACO)*, which trains agents from the human interventions and partial demonstrations. The method incorporates the human expert in the interaction between agent and environment to ensure safe and efficient exploration. The experiments on safe driving show that the proposed method achieves superior training-time safety, outperforming RL and IL baselines. Besides, it shows a high sample efficiency for rapid learning. The constrained optimization technique is used to prevent the agent from excessively exploiting the human expert, which also decreases the takeover frequency and saves valuable human budget.

One limitation of this work is that the trained agents behave conservatively compared to the agents from RL baselines. Aiming to ensure the training time safety of the copilot system, human expert typically slow the vehicle down to rescue it from risky situations. This makes the agent tend to drive slowly and exhibit behaviors such as frequent yielding in the intersection. In future work, we will explore the possibility of learning more sophisticated skills.

**Acknowledgments** This project was supported by the Centre for Perceptual and Interactive Intelligence (CPII) Ltd under InnoHK supported by the Innovation and Technology Commission.

ETHICS STATEMENT

The proposed Human-AI Copilot Optimization algorithm aims at developing a new human-friendly human-in-the-loop training framework. We successfully increase the level of automation after human-efficient training. We believe this work has a great positive social impact which advances the development of more intelligent AI systems that costs less human burdens.

We employ human subjects to participate in the experiments. Human subjects can stop the experiment if any discomfort happens. No human subjects were harmed in the experiments since we test in the driving simulator. The human subjects earn an hourly salary more than average in our community. Each experiment lasts near one hour. Human participants will rest at least three hours after one experiment. During training and data processing, no personal information is revealed in the collected dataset or the trained agents.

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

APPENDIX

The **demonstrative video** and the **code** of HACO and baselines are provided at: `https://decisionforce.github.io/HACO/`

## A    MAIN THEOREM AND THE PROOF

In this section, we derive the upper bound of the discounted probability of failure of HACO, showing that we can bound the training safety with the guardian.

**Theorem 1** (Upper bound of training risk). *The expected cumulative probability of failure $V_{\pi_b}$ of the behavior policy $\pi_b$ in HACO is bounded by the error rate of the human expert action $\epsilon$, the error rate of the human expert intervention $\kappa$ and the tolerance of the human expert $K'$:*

$$V_{\pi_b} \leq \frac{1}{1-\gamma}[\epsilon + \kappa + \frac{\gamma\epsilon^2}{1-\gamma}K'],$$

*wherein $K' = \max_s K(s) = \max_s \int_{a \in \mathcal{A}_h(s)} da \geq 0$ is called human expert tolerance.*

The human expert tolerance $K'$ will becomes larger, if human relieves its intervention and allows the agent to explore the environment more freely.

The proof is given as follows.

***Notations.*** Before starting, we firstly recap and describe the notations. In HACO, a human subject copilots with the learning agent. The agent's policy is $\pi_n$, the human's policy is $\pi_h$. Both policies produces action in the bounded action space $\mathcal{A} \in \mathbb{R}^{|\mathcal{A}|}$. The human expert decides to intervention under certain state and agent's action $a_n$. The human intervention is denoted by a Boolean function: $I(s, a)$. The mixed behavior policy $\pi_b$ that produces the real actions applied to the environment is denoted as:

$$\pi_b(a|s) = \pi_n(a|s)(1 - I(s,a)) + \pi_h(a|s)G(s), \tag{9}$$

wherein $G(s) = \int_{a' \in \mathcal{A}} I(s, a')\pi_n(a'|s)da'$ is a function which denotes the probability of choosing an action that will be rejected by the human.

Therefore, at a given state, we can split the action space into two parts: where intervention will happen or will not happen if the agent sample action in it. We denote the *confident action space* as:

$$\mathcal{A}_h(s) = \{a : I(a|s) \text{ is False}\}. \tag{10}$$

The confident action space contains the actions that will not be rejected by human expert at state $s$.

We also define the ground-truth indicator $C^{gt}$ denoting whether the action will lead to unsafe state. This unsafe state is determined by the environment and is not revealed to learning algorithm:

$$C^{gt}(s, a) = \begin{cases} 1, & \text{if } s' = \mathcal{P}(s'|s, a) \text{ is an unsafe state,} \\ 0, & \text{otherwise.} \end{cases} \tag{11}$$

Therefore, at a given state $s$ the *step-wise probability of failure* for arbitrary policy $\pi$ is:

$$\mathbb{E}_{a \sim \pi(\cdot|s)} C^{gt}(s, a) \in [0, 1]. \tag{12}$$

Now we denote the *cumulative discounted probability of failure* as:

$$V_\pi(s_t) = \mathbb{E}_{\tau \sim \pi} \sum_{t'=t} \gamma^{t'-t} C^{gt}(s_{t'}, a_{t'}), \tag{13}$$

which counts for the chance of entering dangerous states in current time step as well as in future trajectories deduced by the policy $\pi$. We use $V_{\pi_h} = \mathbb{E}_{\tau \sim \pi_h} V_{\pi_h}(s_0)$ to denote the expected cumulative discounted probability of failure of the human. Following the same definition as $V_{\pi_h}$, we can also write the expected cumulative discounted probability of failure of the behavior policy as: $V_{\pi_b} = \mathbb{E}_{\tau \sim \pi_b} V_{\pi_b}(s_0) = \mathbb{E}_{\pi_b} \sum_{t=0} \gamma^t C^{gt}(s_t, a_t)$.

***Assumption.*** Now we introduce two important assumptions on the human expert.

**Assumption 1** (Error rate of human action). *For all states, the step-wise probability of that the human expert produces an unsafe action is bounded by a small value $\epsilon < 1$:*

$$\mathbb{E}_{a \sim \pi_h(\cdot|s)} C^{gt}(s,a) \leq \epsilon. \tag{14}$$

**Assumption 2** (Error rate of human intervention). *For all states, the step-wise probability of that the human expert does not intervene when agent produces an unsafe action is bounded by a small value $\kappa < 1$:*

$$\int_{a \in \mathcal{A}} [1 - I(s,a)]C^{gt}(s,a)da = \int_{a \in \mathcal{A}_h(s)} C^{gt}(s,a)da \leq \kappa. \tag{15}$$

These two assumptions does not impose any constrain on the structure of the human expert policy.

***Lemmas.*** We propose several useful lemmas and the correspondent proofs, which are used in the main theorem.

**Lemma 2** (The performance difference lemma).

$$V_{\pi_b} = V_{\pi_h} + \frac{1}{1 - \gamma} \mathbb{E}_{s \sim P_{\pi_b}} \mathbb{E}_{a \sim \pi_b} [A_{\pi_h}(s,a)]. \tag{16}$$

Here the $P_{\pi_b}$ means the states are subject to the marginal state distribution deduced by the behavior policy $\pi_b$. $A_{\pi_h}(s,a)$ is the advantage of the expert in current state action pair: $A_{\pi_h}(s,a) = C^{gt}(s,a) + \gamma V_{\pi_h}(s') - V_{\pi_h}(s)$ and $s' = \mathcal{P}(s,a)$ is the next state. This lemma is proposed and proved by Kakade & Langford (2002) and is useful to show the behavior policy's safety. In the original proposition, the $V$ and $A$ represents the expected discounted return and advantage w.r.t. the reward, respectively. However, we replace the reward with the indicator $C^{gt}$ so that the value function $V_{\pi_b}$ and $V_{\pi_h}$ presenting the expected cumulative failure probability.

**Lemma 3.** *The cumulative probability of failure of the expert $V_{\pi_h}(s)$ is bounded for all state:*

$$V_{\pi_h}(s) \leq \frac{\epsilon}{1 - \gamma}$$

*Proof.* Following Assumption 1:

$$V_{\pi_h}(s_t) = \mathbb{E}_{\pi_h} \left[ \sum_{t'=t}^{\infty} \gamma^{t'-t} C^{gt}(s_{t'}, a_{t'}) \right] = \sum_{t'=t}^{\infty} \gamma^{t'-t} \mathbb{E}_{\pi_h} [C^{gt}(s_{t'}, a_{t'})] \leq \sum_{t'=t}^{\infty} \gamma^{t'-t} \epsilon = \frac{\epsilon}{1 - \gamma} \quad (17)$$

$\square$

***Theorem.*** We introduce the main theorem of this work above, which shows that the training safety is related to the error rate on action $\epsilon$ and the error rate on intervention $\kappa$ of the human expert. The proof is given as follows.

*Proof.* We firstly decompose the advantage by splitting the behavior policy:

$$\mathbb{E}_{a \sim \pi_b(\cdot|s)} A_{\pi_h}(s,a) = \int_{a \in \mathcal{A}} \pi_b(a|s) A_{\pi_h}(s,a)$$

$$= \int_{a \in \mathcal{A}} \{\pi_n(a|s)(1 - I(s,a))A_{\pi_h}(s,a) + \pi_h(a|s)G(s)A_{\pi_h}(s,a)\}da \tag{18}$$

$$= \int_{a \in \mathcal{A}_h(s)} [\pi_n(a|s)A_{\pi_h}(s,a)]da + G(s) \mathbb{E}_{a \sim \pi_h} [A_{\pi_h}(s,a)].$$

The second term is equal to zero according to the definition of advantage. We only need to compute the first term. We expand the advantage into detailed form, we have:

$$
\begin{aligned}
\mathop{\mathbb{E}}_{a\sim\pi_b(\cdot|s)} A_{\pi_h}(s,a) &= \int_{a\in\mathcal{A}_h(s)} [\pi_n(a|s)A_{\pi_h}(s,a)]da \\
&= \int_{a\in\mathcal{A}_h(s)} \pi_n(a|s)[C^{gt}(s,a)+\gamma V_{\pi_h}(s')-V_{\pi_h}(s)]da \\
&= \underbrace{\int_{a\in\mathcal{A}_h(s)} \pi(a|s)C^{gt}(s,a)da}_{(a)} + \underbrace{\gamma\int_{a\in\mathcal{A}_h(s)} \pi(a|s)V_{\pi_h}(s')da}_{(b)} - \underbrace{\int_{a\in\mathcal{A}_h(s)} \pi(a|s)V_{\pi_h}(s)da}_{(c)} .
\end{aligned}
\tag{19}
$$

Following the Assumption 1, the term (a) can be bounded as:

$$
\int_{a\in\mathcal{A}_h(s)} \pi(a|s)C^{gt}(s,a)da \le \int_{a\in\mathcal{A}_h(s)} C^{gt}(s,a)da \le \kappa. \tag{20}
$$

Following the Lemma 3, the term (b) can be written as:

$$
\gamma\int_{a\in\mathcal{A}_h(s)} \pi(a|s)V_{\pi_h}(s')da \le \gamma\int_{a\in\mathcal{A}_h(s)} V_{\pi_h}(s')da \le \frac{\gamma\epsilon}{1-\gamma}\int_{a\in\mathcal{A}_h(s)} da = \frac{\gamma\epsilon}{1-\gamma}K(s), \tag{21}
$$

wherein $K(s) = \int_{a\in\mathcal{A}_h(s)} da$ denoting the area of human-preferable region in the action space. It is a function related to the human expert and state.

The term (c) is always non-negative, so after applying the minus to term (c) the negative term will always be $\le 0$.

Aggregating the upper bounds of three terms, we have the bound on the advantage:

$$
\mathop{\mathbb{E}}_{a\sim\pi_b} A_{\pi_h}(s,a) \le \kappa + \frac{\gamma\epsilon}{1-\gamma}K(s) \tag{22}
$$

Now we put Eq. 22 as well as Lemma 3 into the performance difference lemma (Lemma 2), we have:

$$
\begin{aligned}
V_{\pi_b} &= V_{\pi_h} + \frac{1}{1-\gamma}\mathop{\mathbb{E}}_{s\sim P_{\pi_b}}\mathop{\mathbb{E}}_{a\sim\pi_b} [A_{\pi_h}(s,a)] \\
&\le \frac{\epsilon}{1-\gamma} + \frac{1}{1-\gamma}[\kappa + \frac{\gamma\epsilon}{1-\gamma}\max_s K(s)]] \\
&= \frac{1}{1-\gamma}[\epsilon + \kappa + \frac{\gamma\epsilon^2}{1-\gamma}K'],
\end{aligned}
\tag{23}
$$

wherein $K' = \max_s K(s) = \max_s \int_{a\in\mathcal{A}_h(s)} da \ge 0$ is correlated to the tolerance of the expert. If the human expert has higher tolerance then $K'$ should be greater.

Now we have proved the upper bound of the discounted probability of failure for the behavior policy in our method.

$\square$

# B VISUALIZATION OF LEARNED PROXY VALUE FUNCTION

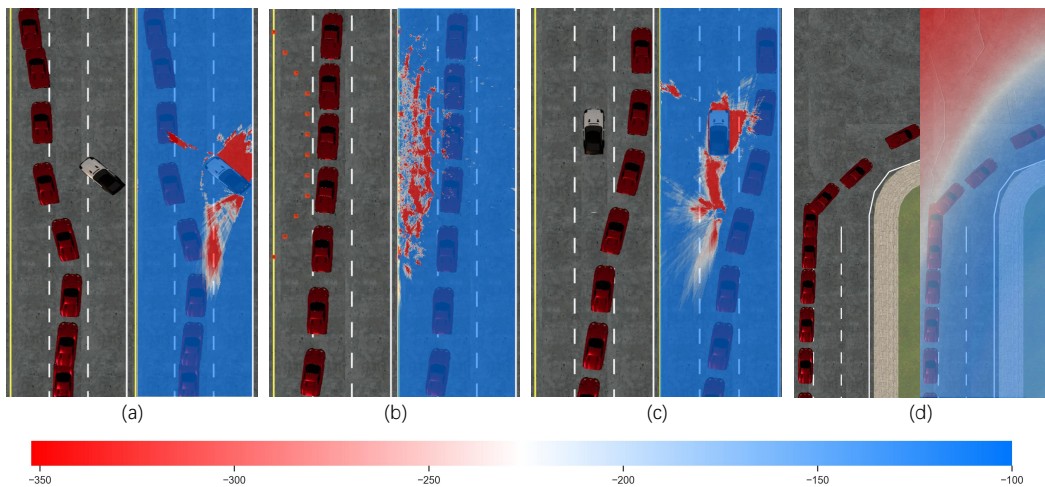

Figure 4: Visualization of proxy Q value learned by HACO.

To understand how well the proxy value function learns, we visualize 4 common scenarios in 4 pairs of figures as shown above. The left sub-figure of each pair shows a top-down view of a driving scenario, where a sequence of snapshots of the control vehicle is plotted, showing its trajectory. The right sub-figure of each pair overlaps the heatmap of proxy values in the top-down image. We manually position the vehicle in different location in the map and query the policy to get action and run the proxy Q function to get the value $Q(s, a)$. Region in red color indicates the proxy value is low if the agent locates there and vice versa.

In Fig. 4(a), the agent performs a lane change behavior to avoid potential collisions with a traffic vehicle which is merging into the middle lane. The region near the traffic vehicle has extremely low values and thus the agent has small probability to enter this area.

In Fig. 4(b), traffic cones spread in the left lane. The agent learns to avoid crashes and the proxy value heatmap shows a large region of low values.

As shown in the trajectory in Fig. 4(c), though the agent can choose to bypass the traffic vehicle in both left-hand side or right-hand side, it chooses the right-hand side. The heatmap shows that much higher proxy Q value is produced on right bypassing path compared to left path. This behavior resembles the preference of human who prefers right-hand side detour.

In addition, in some ares where paths boundary is ambiguous such as the intersection, the agent manages to learn a virtual boundary in the proxy Q space for efficiently passing these areas, as shown in the Fig. 4(d).

The proxy Q value distribution shown in this section not only explains the avoidance behaviors, but also serves as a good indicator for the learned human preference.

## C    DETAILS OF HUMAN-IN-THE-LOOP BASELINES

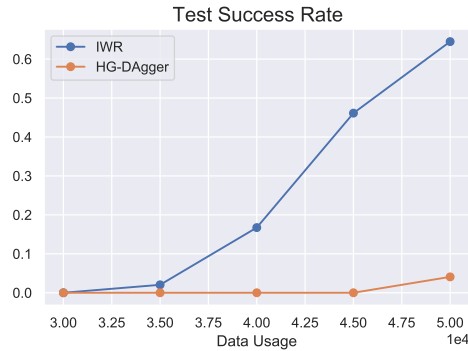

Figure 5: Detailed learning curves of human-in-the-loop baselines.

We benchmark the performance of two human-in-the-loop methods HG-DAgger (Kelly et al., 2019) and IWR (Mandlekar et al., 2020). Both methods require warming up through behavior cloning on a pre-collected dataset. In practice, we find that using 10K or 20K steps of human collected data is not enough to initialize the policy with basic driving skills. Therefore, we use the pre-collected human dataset containing 30K transitions to warm up the policies. After warming up, HG-DAgger and IWR then aggregate human intervention data to the training buffer and conduct behavior cloning again to update policy for 4 epochs. In each epoch the human-AI system collects 5000 transitions. The above figure shows the learning curves of IWR and HG-DAgger. As discussed in the main body of paper, we credit the success of IWR to the re-weighting of human intervention data, which is not emphasized in HG-DAgger.

## D    MORE ZOOM-IN PLOT OF THE LEARNING CURVES

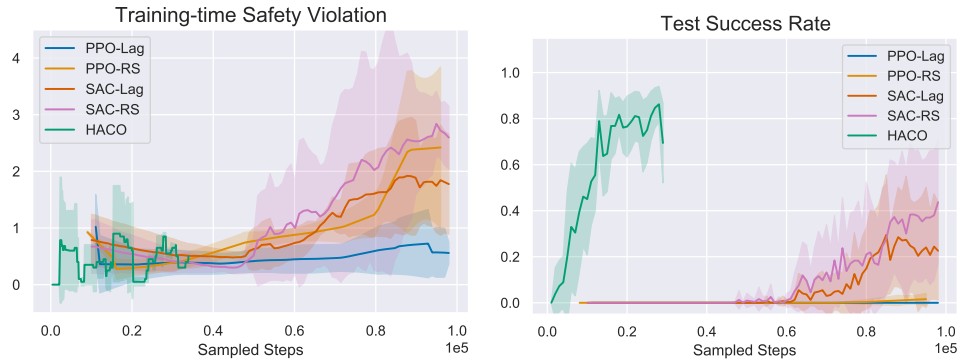

Figure 6: The zoomed in learning curves of RL baselines and HACO.

The above figures present the zoomed in learning curves of RL baselines and HACO, showing the superior sample efficiency of HACO compared to RL baselines.

# E HYPER-PARAMETERS

Table 4: HACO

| Hyper-parameter | Value |
|---|---|
| Discounted Factor $\gamma$ | 0.99 |
| $\tau$ for Target Network Update | 0.005 |
| Learning Rate | 0.0001 |
| Environmental Horizon $T$ | 1000 |
| Steps before Learning Start | 100 |
| Steps per Iteration | 100 |
| Train Batch Size | 1024 |
| CQL Loss Temperature | 10.0 |
| Target Entropy | 2.0 |

Table 5: PPO/PPO-Lag

| Hyper-parameter | Value |
|---|---|
| KL Coefficient | 0.2 |
| $\lambda$ for GAE (Schulman et al., 2018) | 0.95 |
| Discounted Factor $\gamma$ | 0.99 |
| Number of SGD epochs | 20 |
| Train Batch Size | 4000 |
| SGD mini batch size | 100 |
| Learning Rate | 0.00005 |
| Clip Parameter $\epsilon$ | 0.2 |
| Cost Limit for PPO-Lag | 1 |

Table 6: SAC/SAC-Lag/CQL

| Hyper-parameter | Value |
|---|---|
| Discounted Factor $\gamma$ | 0.99 |
| $\tau$ for target network update | 0.005 |
| Learning Rate | 0.0001 |
| Environmental horizon $T$ | 1500 |
| Steps before Learning start | 10000 |
| Cost Limit for SAC-Lag | 1 |
| BC iterations for CQL | 200000 |
| CQL Loss Temperature $\beta$ | 5 |
| Min Q Weight Multiplier | 0.2 |

Table 7: BC

| Hyper-parameter | Value |
|---|---|
| Dataset Size | 36,000 |
| SGD Batch Size | 32 |
| SGD Epoch | 200000 |
| Learning Rate | 0.0001 |

Table 8: CPO

| Hyper-parameter | Value |
|---|---|
| KL Coefficient | 0.2 |
| $\lambda$ for GAE (Schulman et al., 2018) | 0.95 |
| Discounted Factor $\gamma$ | 0.99 |
| Number of SGD epochs | 20 |
| Train Batch Size | 8000 |
| SGD mini batch size | 100 |
| Learning Rate | 0.00005 |
| Clip Parameter $\epsilon$ | 0.2 |
| Cost Limit | 1 |

Table 9: GAIL

| Hyper-parameter | Value |
|---|---|
| Dataset Size | 36,000 |
| SGD Batch Size | 64 |
| Sample Batch Size | 12800 |
| Generator Learning Rate | 0.0001 |
| Discriminator Learning Rate | 0.005 |
| Generator Optimization Epoch | 5 |
| Discriminator Optimization Epoch | 2000 |
| Clip Parameter $\epsilon$ | 0.2 |

Table 10: HG-DAgger

| Hyper-parameter | Value |
|---|---|
| Initializing dataset size | 30K |
| Number of data aggregation epoch | 4 |
| Interactions per round | 5000 |
| SGD batch size | 256 |
| Learning rate | 0.0004 |

Table 11: IWR

| Hyper-parameter | Value |
|---|---|
| Initializing dataset size | 30K |
| Number of data aggregation epoch | 4 |
| Interactions per round | 5000 |
| SGD batch size | 256 |
| Learning rate | 0.0004 |
| Re-weight data distribution | True |

