# OpenReview forum: "Efficient Learning of Safe Driving Policy via Human-AI Copilot Optimization"
_ICLR.cc/2022/Conference — ICLR 2022 Poster_

### Official Review · Reviewer_cwDu · 2021-11-02

**Correctness:** 3
**Technical Novelty And Significance:** 3
**Empirical Novelty And Significance:** 4
**Recommendation:** 8
**Confidence:** 3

**Main Review:**

=== Strengths ===

+ The paper is well written and the presentation is easy to follow.
+ The overall idea and the human-in-the-loop setup are nice and suit very well with real-world applications.
+ The presented approach is simple and effective, and has strong experimental performance compared to IL and is on-par with RL, and does not require extra reward function. It is much more sample efficient than the common RL approaches.
+ The paper also presents a novel driving simulator with procedurally generated maps and active agents. The authors mention that the simulator will be released, which is a big plus.

=== Weaknesses ===

- I like the analysis that justifies the design choice of equation 6, but I think the proposed solution is slightly overfitting to the environment (for example, steering which the environment uses), since in general action space does not necessarily form a metric space.
- There is a typo in equation 3.
- The authors mention the method builds upon CQL, however judging from equation 5 it does not seem to strictly follow CQL, but looks more like a standard Q learning loss. I would appreciate it if the authors can clarify this.


**Summary Of The Paper:**

This paper presents HACO, a human-in-the-loop learning algorithm that aims to learn imitative driving policy while minimizing the number of human interventions. HACO builds on CQL and operates under the no-reward assumption. HACO learns a proxy action-value function by penalizing the policy’s action and maximizing during human interventions. It additionally adds an entropy term to encourage exploration. The policy trains by maximizing the proxy action-value, and penalizing an accumulative intervention cost, computed using the cosine difference between the human and the policy actions. HACO is evaluated in a closed-loop driving simulator. HACO outperforms the selected imitation and offline RL baseline and is on par with RL methods which have access to environment rewards. It is also orders of magnitude more sample efficient than standard RL methods.

**Summary Of The Review:**

This paper presents a simple yet very effective human-in-the-loop learning algorithm. The idea of minimizing human intervention is nice, and the presented approach has strong experimental performance. The novel simulator on which the presented approach is evaluated will be released, which is another plus. I recommend acceptance for this paper.

---

> ### Author Response · Authors · 2021-11-12
> **Response to Reviewer cwDU**
>
> Thank you for the comments! We are happy that you like this work and the simulator! Please find the responses below.
>
> ---
>
> *Q: Cosine similarity is overfitted to the environment, since in general action space does not necessarily form a metric space.*
>
> A: Indeed. The cosine similarity is proposed to avoid the cases when the human interventions are incorrectly applied to delayed time steps due to latency between perception and  feedback. We believe there must exist a better solution to tackle the human feedback delay problem elegantly.
>
> ---
>
> *Q: Equation 5 does not seem to follow CQL but instead like a Q learning loss.*
>
> A: The difference between Equation 5 in our work and the CQL loss is that we add a mask to apply the CQL loss only in the partial demonstration provided when human intervention occurs. The first part of Equation 5 is a Q learning loss which approximates the TD-target and the second term is the masked CQL loss.

---

> > ### Comment · Reviewer_cwDu · 2021-11-12
> > **Thanks for the clarification**
> >
> > Thank you for the clarification, I now see how equation 5 resembles the CQL loss.

---

### Official Review · Reviewer_Mao2 · 2021-11-02

**Correctness:** 4
**Technical Novelty And Significance:** 3
**Empirical Novelty And Significance:** 2
**Recommendation:** 6
**Confidence:** 3

**Main Review:**

Strengths:
1. This paper is well written with clear logic and goals. The experiments are comprehensive. The proposed method is compared with several types of baselines including RL, safe RL, offline RL, and IL.
2. The method proposed for co-learning and intervention minimization is solid.

Weakness:
1. The description of the method can be improved. There seem to be some errors in the equations. For example, in equation (3), the two Q(s, \hat{a}) are the same.
2. I am not sure how to determine the convergence of the algorithm. Why HACO has much fewer steps than the baselines. Especially during the testing phase, why HACO also has fewer steps?
3. The main part of the methodology is based on batch RL. The adding of intervention minimization seems to be a marginal contribution.
4. The driving tasks are considered to be too easy.

**Summary Of The Paper:**

This paper proposed a method for driving policy learning based on human-AI copilot. The algorithm learns from human interventions and also tries to minimize the total efforts of human intervention. Comprehensive experiments and comparisons with multiple baselines show that the proposed algorithm can achieve high sample efficiency and reduce unsafe events. The contributions are in the design of the copilot learning method.

**Summary Of The Review:**

The paper is solid and clear, with enough experimental support. I think it is higher than the bar.

---

> ### Author Response · Authors · 2021-11-12
> **Response to Reviewer Mao2**
>
> Thanks for your review! Please find the detailed responses below.
>
> ---
>
> *Q: I am not sure how to determine the convergence of the algorithm. Why HACO has much fewer steps than the baselines. Especially during the testing phase, why HACO also has fewer steps?*
>
> A: First, there is no explicit test phase. In Table 1 and Figure 3, we run the intermediate checkpoints to see their performance in test environments after the training of HACO.
>
> Second, we ask the human expert to run a fixed time budget (50 minutes, 30000 steps). In our preliminary experiment, we find that the takeover rate reduces drastically and finally reaches a plateau, meanwhile the human subjects report that they are confident  that the agent already mastered how to drive. Therefore, we empirically set the total human budget to 50 minutes of training time.
>
> Third, we hypothesize the high sample efficiency is because the learning policy in HACO is updated more frequently, compared to DAgger-like methods. More importantly, we consider the learning on a trajectory basis, instead of optimizing agents to imitate exact actions at each time step. Adopting the Q learning as backbone indicates we incentivize the agent to choose an action that can bring potential return in future trajectory, instead of only mimicking the expert’s behaviors at each step.
>
> ---
>
> *Q: The main part of the methodology is based on batch RL. The addition of intervention minimization seems to be a marginal contribution.*
>
> A: In fact, the major contribution of this work is the human-AI copilot framework and our integrated solution. The Offline RL method is applied in HACO as a natural solution to our trajectory-based problem formulation, and the takeover minimization addresses the **issue of the agent overusing and wasting human interventions**.
>
> The takeover minimization is a necessary component empowering HACO to solve tasks without human intervention. Since the proxy-Q value estimates the accumulative expected return on trajectories sampled by the human-AI mixed policy. The takeover minimization is implemented to minimize the ratio of human intervention and thus the agent can finish tasks without human supervision. As already presented in the ablation study in Section 4.3, we find that removing the intervention minimization breaks HACO completely.
>
> ---
>
> *Q: The driving tasks are considered to be too easy.*
>
> A: We believe the environment is not easy. As you can find in the “Training Data Usage” column of Table 1, the powerful baseline SAC takes 1M steps to master driving. Other powerful baselines such as BC, GAIL, CQL using human demonstrations can not even achieve satisfactory results. Besides, we split the training and test sets containing diverse driving scenarios and report the test performance in the test set. The agent is supposed to acquire complex and safe driving skills to reach destination and avoid crashes and generalizability to unseen scenarios, which is a hard task.

---

### Official Review · Reviewer_ohEo · 2021-11-02

**Correctness:** 3
**Technical Novelty And Significance:** 3
**Empirical Novelty And Significance:** 3
**Recommendation:** 6
**Confidence:** 3

**Main Review:**

Overall, I enjoyed reading this paper and found its results convincing. However, there is one missing experimental comparison that I would like to see before I can recommend an accept: prior work on human-in-the-loop imitation learning [2] proposes the Intervention-Weighted Regression (IWR) method, which tackles the same problem under the same assumptions as HACO. While HACO directly penalizes the agent for triggering human interventions, IWR takes a different approach based on dataset balancing. To judge the relative performance and contribution of HACO, it would be helpful to implement and run IWR on the driving task in Table 1 and Figures 3-4.

Missing related work:
 1. [Learning from Interventions](http://www.roboticsproceedings.org/rss16/p055.pdf)
 2. [Human-in-the-Loop Imitation Learning using Remote Teleoperation](https://arxiv.org/abs/2012.06733)

Minor comments:
 - Is there a typo in Equation 3? As written, the objective evaluates to zero. I'm also confused as to why offline RL is necessary to fit the Q-function, since we do not have a reward function in this setting and instead have partial demonstrations. Can't we just use behavioral cloning to fit an imitation policy?
 - Why does the training cost go *up* over time in the left panel of Figure 3?

Update
----
Thank you to the authors for adding the comparison to IWR. I have increased my score.


**Summary Of The Paper:**

The paper proposes HACO, a human-in-the-loop reinforcement learning method that safely trains an agent to imitate expert behavior while minimizing the number of expert interventions required. The key idea is to have a human watch over the agent (e.g., in a simulated driving environment), and take control whenever the agent enters unsafe states. HACO uses offline RL to train the agent to imitate the actions taken by the human during these interventions. To discourage the agent from intentionally visiting unsafe states in order to trigger human interventions, HACO also assigns a negative reward to the state transitions preceding a human intervention. Experiments with human participants in a simulated driving task show that HACO trains the agent to achieve higher success rates (in a test environment, without a human in the loop) than baseline methods based on imitation learning and offline RL, while requiring less training data and incurring a lower cumulative training cost.

**Summary Of The Review:**

Missing crucial comparison to prior method

---

> ### Author Response · Authors · 2021-11-12
> **Response to Reviewer ohEo**
>
> Thanks for your review. Please find our responses below.
>
> ---
>
> *Q: Why offline RL is necessary to fit the Q-function. Can’t we just use behavior cloning to fit an imitation policy?*
>
> A: You pointed out the most novel part of this work. We consider the learning on a trajectory basis, instead of optimizing agents to imitate exact actions at each time step. Adopting the Q learning as backbone indicates we incentivize the agent to choose an action that can bring potential return in future trajectory, instead of only mimicking the expert’s behaviors at each step.
>
> We consider the human-AI copilot problem as a guided reinforcement learning problem where the agent learns from exploration instead of an imitation learning problem. Techniques from offline RL are more suitable in this setting.
>
> We already conducted an experiment in the paper showing the performance of Behavior Cloning based on the human expert trajectories, shown in Table 1. We also experiment new baselines using the collected trajectories during human-AI copilot to train behavior cloning agents. Please refer to the Common Response.
>
> ---
>
> *Q: You can run the baseline Intervention-Weighted Regression (IWR).*
>
> A: Yes! Thanks for the suggestion. We add this baseline experiment and find IWR is a strong baseline but our method still outperforms it in terms of test-time success rate and test-time safety. Please refer to the Common Response for details.
>
> ---
>
> *Q: You should add some related works into the discussion.*
>
> A: Yes, thanks for pointing out those insightful works. We add them in the related work section.

---

### Official Review · Reviewer_irh4 · 2021-11-04

**Correctness:** 3
**Technical Novelty And Significance:** 3
**Empirical Novelty And Significance:** 3
**Recommendation:** 6
**Confidence:** 3

**Main Review:**

### Quality

The paper's results are quite interesting, and the experiments – as far as I can tell – seem well executed. I particularly appreciated the additional ablations provided in Table 2. The method seems relatively well motivated. The method proposed seems to do very well relative to the chosen baselines along the performance metrics examined.

### Clarity & Limitations
Stylistically the paper could use more editing. There are various typos (some below) and many phrasings that could be improved.

There are also various things that could be improved in the results' clarity, or which seem to constitute limitations:
- Who is intervening? 3 experts – how were they selected? While the training is happening, what are the experts doing? "The whole training takes about 50 minutes." Is the expert providing actions for the whole time?
- "The main experiments of HACO repeat 3 times." -> what does this mean? With each human expert, you perform one training run?
- Generally would be helpful to give more intuition as to why IL is being outperformed with much less data?
- "We split the driving scenes into the training set and test set with 50 different scenes in each set. At the beginning of each episode, a scene in the training or test set is randomly selected." -> The latter part of this phrase seems to suggest that you're training on the test set?
- I found it confusing how in Section 3.3, different costs and rewards were defined one by one, without a high-level motivation. Before describing each one, I would state clearly that you are considering _various different metrics of performance_: "Test Return", "Test Cost" (which is different than just negative returns!), and "Test Success Rate". I would potentially propose changing the name from "Cost" to "Safety Violations", and from "Success Rate" to "Goal completion" to make it easier to disambiguate what the motivation between using these multiple metrics is. As currently phrased, they all seem synonymous with "reward" (or the opposite of it) on a first read.
- In Section 3.3, I would switch to introduce cosine similarity first, and then say that in ideal case this is not necessary. I found presenting things in the current order more confusing.
- The sentence "Frequent querying might exhaust the human expert and brings tremendous cognitive costs (Zhang et al., 2021)" is almost verbatim repeated both in section 3 and in section 2. I would remove it from 2.
- "Under the protection from the human expert, HACO yields only 30.14 total training cost in the whole training process, an order of magnitude better than the other baselines," -> if I'm reading Table 1 correctly, it's essentially 2 orders of magnitude better? The next best value seems to be 1840
- Figure 3: it might be useful to have zoomed in (on the x axis) training curves in the appendix. Are the shadings of the training curves standard errors?

A limitation seems to be that of implicit assumptions made by the method: using CQL as in eq. (3) is making relatively strong assumptions about the nature of expert intervention. While it seems to work well in practice, the form of eq. (6) also seems quite arbitrary, and it's unclear how one would extend it to other environments (e.g. with discrete action spaces).

Additionally, the lack of human-in-the-loop baselines seems potentially problematic – after all, the method is presented as a improvement in that line of work. The same could be said about the environment – given how promising this method seems to be in this context, it would be very interesting to see how it performs in other environments. While I realize that this is not something that can be addressed in the timeframe of rebuttals, it would significantly strengthen the paper.

Typos:
- "To encourage the exploration in the area that does not _violating_ human intention,"
- "Human-Gated DAgger (HG-DAgger) (Kelly et al., 2019) utilizes an expert to _intervene exploration_ and"
- "Other forms of human participation _includes_ providing human preferences"
- "rewarding the state-action pairs deduced _by human_ and penalizing those _by agent_."
- "The speed reward _is vehicle’s_ current speed"
- "Since we test different approaches in the driving simulator, no injury would _happens_."

### Originality

Although I'm not an expert in this area, as far as I know, this seems like a novel contribution.

**Summary Of The Paper:**

In this work, the authors propose a new algorithm for data-efficient human-in-the-loop learning, Human-AI Copilot Optimization. The main idea is to have experts intervene during training in cases in which unsafe situations arise. The HACO learned policy utilizes a multi-task objective: doing well relative to a learned value function (based on human interventions), keeping an exploratory policy, and keeping human interventions at a minimum. Experimentally, HACO seems to be able to drastically reduce the amount of number of environment training timesteps required to reach basic competencies to the agent in the test environment, while maintaining good task and safety performance.

**Summary Of The Review:**

Overall, the novel method presented by this paper, HACO, seems to be able to greatly outperform safe RL, IL, and Offline RL baselines – leading it to being able to achieve relatively similar safety and performance metrics while incurring in many less unsafe situations during training time, and while using 2 orders of magnitude less data than RL approaches. While some portions of the paper and motivations for the method lack clarity and baselining is limited to non human-in-the-loop methods, the strength of the empirical results still suggests that the submission is potentially worth accepting.

---

> ### Author Response · Authors · 2021-11-12
> **Response to Reviewer irh4**
>
> Thank you very much for providing such a detailed review and suggestions for improving the clarity and quality of our paper! Apart from the paper revision and typo fixing, we have the following responses for your questions.
>
> ---
>
> *Q: How many experts, how are they selected and what are they doing in copilot?*
>
> A: In one experiment, there is only one human expert co-piloting with the agent. Since we repeat three times for the main HACO experiments, we invite three different individuals to serve as the experts. They are ordinary students in our community with average skills to drive in the simulator.
>
> The expert **watches over** the exploration of the learning agent and **occasionally takes over and provides demonstrations**. Therefore the expert only provides actions if s/he decides to take over. In our experiment, in 27.7% of the total 50 minutes of training the expert is providing actions, which is much less than the imitation learning baseline requiring tremendous human demonstration.
>
> ---
>
> *Q: Why IL methods are outperformed by HACO with much less data?*
>
> A: We think this is because we consider the learning on a trajectory basis, instead of optimizing agents to imitate exact actions at each time step. Adopting the Q learning as backbone indicates we incentivize the agent to choose an action that can bring potential return in future trajectory, instead of only mimicking the expert’s behaviors at each step.
>
> ---
>
> *Q: How one could extend HACO to other environments by adapting the CQL and the cosine similarity?*
>
> A: We think CQL technique can be easily adapted to other environments because it does not constrain the form of action space. On the contrary, the form of intervention cost in HACO is indeed couped to the continuous action space.
>
> Since the cosine similarity is proposed to avoid the cases when the human interventions are applied to delayed time steps due to the feedback latency, in discrete action space we can use the (1-p) as the intervention cost, where p is the probability of choosing the human's action by the agent's action distribution.
>
> ---
>
> *Q: Are you training in the test set?*
>
> A: No. We train in the training set containing 50 different maps and deploy the trained agent without human supervision in the test set containing another 50 unseen environments. All maps and traffic flows are unique. Thank you for pointing out this ambiguity. We will present this more clearly in our paper.
>
> ---
>
> *Q: You can add other human-in-the-loop baselines.*
>
> A: Thanks for the suggestion, please find new human-in-the-loop baselines in the Common Response.
>
> ---
>
> *Q: You can rephrase the terms of evaluative metrics, the introduction of cosine similarity, etc.*
>
> A: Thanks for your suggestions! We already revised the figures and phrasing in the submission to make it clearer.

---

> > ### Comment · Reviewer_irh4 · 2021-11-15
> > **Additional Questions**
> >
> > Thanks for your replies!
> >
> > Two further follow-up questions are below:
> > - You say the experts are watching the exploration of the learning agent. Is the learning agent acting in real-time (rather than as fast as possible, as allowed by the simulator, as is usually done in RL training)?
> > - It would be nice to state more clearly what the implicit assumptions of using CQL as in eq. (3) are about the nature of expert intervention.

---

> > > ### Author Response · Authors · 2021-11-15
> > > **Follow-up Response to Reviewer irh4**
> > >
> > > Thanks for your quick replies! Please find the response below:
> > >
> > > ---
> > >
> > > *Q1: You say the experts are watching the exploration of the learning agent. Is the learning agent acting in real-time (rather than as fast as possible, as allowed by the simulator, as is usually done in RL training)?*
> > >
> > > A: Yes, as shown in [our anonymized demo video](https://drive.google.com/file/d/1g6Fr90vc-uJhxTlT0Y8eIFFKaz3EAcX-/view) and the statement in Section 4.1 Experimental Settings, the human-AI copilot system is running in real-time.
> > >
> > > In the learning environment, each step lasts 0.1s simulation time. The learning and execution in HACO is efficient enough so that the system can run faster than 10FPS and in real-time.
> > >
> > > ---
> > >
> > > *Q2: It would be nice to state more clearly what the implicit assumptions of using CQL as in eq. (3) are about the nature of expert intervention.*
> > >
> > > A: We assume that **actions from human-provided partial demonstrations can bring the higher accumulated return**. Thus we increase the proxy Q value of human-selected actions and suppress the value of agent-selected actions in the CQL loss. As a result, human knowledge and preference about the task can be injected to the proxy value function.
> > >
> > > To better justify this assumption and examine the correlation between the proxy Q values and the agent behaviors, we visualize the distribution of proxy Q values in Appendix Section B in the revised paper, and provide detailed case studies to analyze how the proxy value Q distribution influences the agent trajectories in 4 common traffic scenarios. The conclusion is that the proxy Q value can successfully encode human preference and guide the learning of agents.

---

> ### Author Response · Authors · 2021-11-17
> **Any further comment?**
>
> Dear Reviewer irh4
>
> Thank you so much for your effort in reviewing this submission! We have tried our best to address your concerns above the reply. There are also extra human-in-the-loop experiments in the common response area. Would you mind taking a look and letting us know what you think?
>
> Feel free to let us know if there is anything unclear or so. We are happy to clarify them.
>
> Best,
> Authors

---

### Author Response · Authors · 2021-11-12
**Common Response to Reviewers**

We thank all reviewers for the insightful reviews. We revised our submission and the major modification is highlighted by red color. Apart from fixing the typos, we:

1. **Add two human-in-the-loop baselines: IWR and HG-Dagger.**
2. Fix the Equation 3 with the correct form of CQL loss.
3. Move the training safety guarantee from a separate external file into the attached Appendix for convenience of the reader.
4. Rename metrics to make them easier to understand (cost -> safety violations).
5. Add visualization of the proxy Q values in the Appendix.

Apart from the specific responses to each reviewer, here we address the common issues raised in the reviews on the paper clarity and human-in-the-loop baselines.

---

*Q: Can you compare relevant human-in-the-loop baselines such as IWR?*

A: We conducted the experiment of IWR (Mandlekar et al., 2021) and HG-Dagger (Kelly et al., 2019). The agents warm up through behavior cloning on a pre-collected dataset containing 30,000 steps of human demonstration. Both methods then append 5,000 steps of human intervention data to the training buffer and do behavior cloning to update policy in each epoch. We conduct 4 epochs of training.

We added the details and results in the revised paper. Here we summarize the experiment result:

|             | Test Episodic Reward | Test Safety Violations | Test Success Rate |
|-------------|----------------------|------------------------|-------------------|
| IWR         | 299.78               | 3.39                   | 0.64              |
| HG-Dagger   | 111.87               | 2.38                   | 0.04              |
| HACO (ours) | **349.25**        | **0.79**             | **0.83**       |


We find that IWR outperforms HG-Dagger and it is a strong baseline, but our method still outperforms IWR in terms of test-time success rate and test-time safety.

Reference:

(Mandlekar et al., 2021) Human-in-the-Loop Imitation Learning using Remote Teleoperation

(Kelly et al., 2019) HG-DAgger: Interactive Imitation Learning with Human Experts

---

### Author Response · Authors · 2021-12-02
**Experimenting HACO in CARLA simulator**

We run HACO in the CARLA simulator. **It can train a driving agent in 10 minutes with only 8,000 transitions in the environment.** Please check the video here:

https://youtu.be/a1uiyHFTRSA

---

Concretely, we train HACO (with a human expert) and PPO in the town 1 of CARLA simulator and report the test performance in the town 2 in the following table.

| Algorithm | Test Safety Violation | Test Episodic Reward | Test Success Rate | Data Utilization (steps) |
| --------- | ----------------- | -------------------- | ----------------- | ---------------- |
| PPO       | 80.84             | 1591.00              | 0.35              | 500,000          |
| HACO      | 11.84             | 1579.03              | 0.35              | 8,000            |


We use the [CARLA simulator](https://carla.org/). We use the top-down semantic view provided by CARLA as the input and utilize 3 layers CNN as the feature extractor for HACO and baselines. The reward is related to the velocity and the completion of the road (again, please note that HACO does not use reward in training). Due to computing resource limitation, we do not introduce traffic vehicles and other participants in the town.

This experiment shows that the proposed HACO framework can be successfully deployed in the CARLA simulator.

---

### Public Comment · ~Fan_Wang4 · 2022-07-29
**Missing Citations**

We want to kindly call attention to that the following papers are very related to this work but not included in the reference.

Wang, Fan, et al. "Intervention aided reinforcement learning for safe and practical policy optimization in navigation." Conference on Robot Learning. PMLR, 2018.

Wu, Jingda, et al. "Human-in-the-loop deep reinforcement learning with application to autonomous driving." arXiv preprint arXiv:2104.07246 (2021).

---

### Decision · Program_Chairs · 2022-01-20

**Decision:**

Accept (Poster)

**Comment:**

The paper presents a new algorithm for augmenting RL training with human examples, and this is applied to learning safe driving policies. This algorithm is properly tested and compared to other relevant algorithms with favorable results. Reviewers agree that this is good work and that it should be published. Reviewers had multiple questions, which were in my opinion answered satisfactorily by the authors. Notably, the authors ran additional tests against other human-in-the-loop RL algorithms with good results. In sum, this seems to be a solid paper, worth accepting.